# Soil Moisture and Hydrology Projections of the Permafrost Region: A Model Intercomparison

Christian G. Andresen[1,2], David M. Lawrence[3], Cathy J. Wilson[1], A. David McGuire[4], Charles Koven[5], Kevin Schaefer[6], Elchin Jafarov[6,1], Shushi Peng[7], Xiaodong Chen[8], Isabelle Gouttevin[9,10], Eleanor Burke[11], Sarah Chadburn[12], Duoying Ji[13], Guangsheng Chen[14], Daniel Hayes[15], Wenxin Zhang[16,17]

[1]Earth and Environmental Science Division, Los Alamos National Laboratory, Los Alamos, New Mexico, USA
[2]Geography Department, University of Wisconsin Madison, Madison, Wisconsin, USA
[3]National Center for Atmospheric Research, Boulder, Colorado, USA
[4]Institute of Arctic Biology, University of Alaska Fairbanks, Fairbanks, Alaska, USA
[5]Climate and Ecosystem Sciences Division, Lawrence Berkeley National Lab, Berkeley, CA, USA
[6]Institute of Arctic Alpine Research, University of Colorado Boulder, Boulder, Colorado, USA
[7] UJF–Grenoble 1/CNRS, Laboratoire de Glaciologie et Géophysique de l'Environnement (LGGE), Grenoble, France
[8]Department of Civil and Environmental Engineering, University of Washington, Seattle, Washington, USA
[9]IRSTEA-HHLY, Lyon, France.
[10]IRSTEA-ETNA, Grenoble, France.
[11]Met Office Hadley Centre, UK
[12]School of Earth and Environment, University of Leeds, UK
[13]College of Global Change and Earth System Science, Beijing Normal University, China
[14]Environmental Sciences Division, Oak Ridge National Laboratory, Oak Ridge, Tennessee, USA
[15] School of Forest Resources, University of Maine, Maine, USA
[16] Department of Physical Geography and Ecosystem Science, Lund University, Lund, Sweden
[17]Center for Permafrost (CENPERM), Department of Geosciences and Natural Resource Management, University of Copenhagen, Denmark

*Correspondence to*: Christian G. Andresen (candresen@wisc.edu)

**Abstract.** This study investigates and compares soil moisture and hydrology projections of broadly-used land models with permafrost processes and highlights the causes and impacts of permafrost zone soil moisture projections. Climate models project warmer temperatures and increases in precipitation (P) which will intensify evapotranspiration (ET) and runoff in land models. However, this study shows that most models project a long-term drying of the surface soil (0-20cm) for the permafrost region despite increases in the net air-surface water flux (P-ET). Drying is generally explained by infiltration of moisture to deeper soil layers as the active layer deepens or permafrost thaws completely. Although most models agree on drying, the projections vary strongly in magnitude and spatial pattern. Land-models tend to agree with decadal runoff trends but underestimate runoff volume when compared to gauge data across the major Arctic river basins, potentially indicating model structural limitations. Coordinated efforts to address the ongoing challenges presented in this study will help reduce uncertainty in our capability to predict the future Arctic hydrological state and associated land-atmosphere biogeochemical processes across spatial and temporal scales.

## 1. Introduction

Hydrology plays a fundamental role in permafrost landscapes by modulating complex interactions among biogeochemical cycling (Frey and Mcclelland, 2009; Newman et al., 2015; Throckmorton et al., 2015), geomorphology (Grosse et al., 2013; Kanevskiy et al., 2017; Lara et al., 2015; Liljedahl et al., 2016) and ecosystem structure and function (Andresen et al., 2017; Avis et al., 2011; Oberbauer et al., 2007). Permafrost has a strong influence on hydrology by controlling surface and sub-surface distribution,

storage, drainage and routing of water. Permafrost prevents vertical water flow which often leads to
saturated soil conditions in continuous permafrost while confining subsurface flow through perennially-
unfrozen zones (a.k.a. taliks) in discontinuous permafrost (Jafarov et al., 2018; Walvoord and Kurylyk,
2016). However, with the observed (Streletskiy et al., 2008) and predicted (Slater and Lawrence, 2013)
thawing of permafrost, there is a large uncertainty in the future hydrological state of permafrost
landscapes and in the associated responses such as the permafrost carbon-climate feedback.
The timing and magnitude of the permafrost carbon-climate feedback is, in part, governed by changes in
surface hydrology, through the regulation by soil moisture of the form of carbon emissions from thawing
labile soils and microbial decomposition as either $CO_2$ or $CH_4$ (Koven et al., 2015; Schädel et al., 2016;
Schaefer et al., 2011). The impact of soil moisture changes on the permafrost-carbon feedback could be
significant. Lawrence et al. (2015) found that the impact of the soil drying projected in simulations with
the Community Land Model decreased the overall Global Warming Potential of the permafrost carbon-
climate feedback by 50%. This decrease was attributed to a much slower increase in $CH_4$ emissions if
surface soils dry, which is partially compensated for by a stronger increase in $CO_2$ emissions under drier
soil conditions.
Earth System Models project an intensification of the hydrological cycle characterized by a general
increase in the magnitude of water fluxes (e.g. precipitation, evapotranspiration and runoff) in northern
latitudes (Rawlins et al., 2010; Swenson et al., 2012). In addition, intensification of the hydrological cycle
is likely to modify the spatial and temporal patterns of water in the landscape. However, the spatial
variability, timing, and reasons for future changes in hydrology in terrestrial landscapes in the Arctic are
unclear and variability in projections of these features by current terrestrial hydrology applied in the
Arctic have not been well documented. Therefore, there is an urgent need to assess and better understand
hydrology simulations in land models and how differences in process representation affect projections of
permafrost landscapes.
Upgrades in permafrost representation such as freeze and thaw processes in the land component of Earth
System Models have improved understanding of the evolution of hydrology in high northern latitudes.
Particularly, soil thermal dynamics and active layer hydrology upgrades include the effects of unfrozen
water on phase change, insulation by snow (Peng et al., 2015), organic soils (Jafarov, E. and Schaefer,
2016; Lawrence et al., 2008) and hydraulic properties of frozen soils (Swenson et al., 2012). Nonetheless,
large discrepancies in projections remain as the current generation of models substantially differ in soil
thermal dynamics (e.g. Peng *et al* 2015, Wang *et al* 2016). In particular, variability among current
models' simulations of the impact of permafrost thaw on soil water and hydrological states is not well
documented. Therefore, in this study we analyze the output of a collection of widely-used "permafrost-
enabled" land models. These models participated in the Permafrost Carbon Network Model
Intercomparison Project (PCN-MIP) (McGuire et al., 2018, 2016) and contained the state-of the art
representations of soil thermal dynamics in high latitudes at that time. In particular, we assess how
changes in active layer thickness and permafrost thaw influence near-surface soil moisture and hydrology
projections under climate change. In addition, we provide comments on the main gaps and challenges in
permafrost hydrology simulations and highlight the potential implications for the permafrost carbon-
climate feedback.




## 2. Methods

### 2.1 Models and Simulation Protocol

This study assesses a collection of terrestrial simulations from models that participated in the PCN-MIP (McGuire et al., 2018, 2016) (Table 1). The analysis presented here is unique as it focuses on the hydrological component of these models. Table 2 describes the main hydrological characteristics for each model. Additional details on participating models regarding soil thermal properties, snow, soil carbon and forcing trends can be found in previous PCN-MIP studies (e.g. McGuire *et al* 2016, Koven *et al* 2015, Wang *et al* 2016, Peng *et al* 2015). It is important to note that the versions of the models presented in this study are from McGuire *et al* (2016, 2018) and some additional improvements to individual models may have been made since then.

The simulation protocol is described in detail in *McGuire et al.*, (2016, 2018). In brief, models' simulations were conducted from 1960 to 2299, partitioned by historic (1960-2009) and future simulations (2010-2299), where future simulations were forced with a common projected climate derived from a fully coupled climate model simulation (CCSM4) (Gent et al., 2011). Historic atmospheric forcing datasets (Table 1) (e.g. climate, atmospheric $CO_2$, N deposition, disturbance, etc.) and spin-up time were specific to each modeling group. The horizontal resolution ($0.5°$ – $1.25°$) and soil hydrological column configurations (depths ranging from 2 to 47m and 3 to 30 soil layers) also vary across models (Figure 1). We focus on results from simulations forced with climate and $CO_2$ from the Representative Concentration Pathway (RCP) 8.5 scenario, which represents unmitigated, "business as usual" emissions of greenhouse gases. Future simulations were calculated from monthly CCSM4 (Gent et al., 2011) climate anomalies for the Representative Concentration Pathway (RCP 8.5, 2006-2100) and the Extension Concentration Pathway (ECP 8.5, 2101-2299) scenarios, relative to repeating (1996-2005) forcing atmospheric datasets from the different modeling groups (Table 1).

The PCN model intercomparison uses the output from a single Earth System model climate projection and was motivated by a desire to keep the experimental design simple and computationally tractable. Clearly, using just one climate projection does not allow us to explore the impact of the broad range of potential climate outcomes that are seen across the CMIP5 models. Instead, the PCN suite of simulations allows for a relatively controlled analysis of the spread of model responses to a single representative climate trajectory. The selection of CCSM4 as the climate projection model was motivated partly by convenience and also because it was one of the only models that had been run out to the year 2300 at the time of the PCN experiments. Further, as noted in McGuire et al. (2018), CCSM4 late 20th century climate biases in the Arctic were among the lowest across the CMIP5 model archive. It should be noted that the use of a single climate projection means that the results presented here should be viewed as indicative of just one possible permafrost hydrologic trajectory. As we will show, even under this single climate trajectory, the range of hydrologic responses in the models are broad, indicating high structural uncertainty across models with respect to this particular aspect of the Arctic system response to global climate change.

### 2.2 Permafrost and Hydrology Variables Analyzed

Our analysis focused on the permafrost regions in the Northern Hemisphere north of $45^0$N. This
qualitative hydrology comparison was based on the full permafrost domain for each model rather than a
common subset among models in order to fully portray the overall changes in permafrost hydrology for
participating models. For each model, we define a grid cell as containing near-surface permafrost based
on soil temperature where the annual monthly maximum active layer thickness (ALT) is at or less than
the 3m depth layer depending on the model soil configuration (Figure 1) (McGuire et al., 2016; Slater and
Lawrence, 2013). We calculated the depth of max ALT by identifying the underlying annual permafrost
table depth of continuous monthly temperatures $<273.15^{\circ}$k in the top 3 meters or equivalent soil layer
depth (Figure 1). Models with a soil configuration at 3 meters or less (UWVIC, CoLM, JULES, TEM)
follow the same calculation with an exemption for their bottom depth, where soil depth temperature
threshold of $<273.5^{\circ}$k was applied to be considered as permafrost, this was based on soil temperature
trends observed for models with deeper soil depths greater than 3 meters and allows models to have a
ALT of 3 meters when soil configuration is limiting. We assessed how permafrost changes affect near-
surface soil moisture, defined here as the soil water content ($kg/m^2$) of the 0-20 cm soil layer. We focused
on the top 20 cm of the soil column due to its relevance to near-surface biogeochemical processes. We
added the weighted fractions for each depth interval to calculate near-surface soil moisture (0-20cm) to
account for the differences in the vertical resolution of the soil grid cells among models (Figure 1). To
better understand the causes and consequences of changes in soil moisture, we examined several principal
hydrology variables including evapotranspiration (ET), runoff (R; surface and sub-surface) and
precipitation (P; snow and rain). Representation of ET, R and soil hydrology varies across participating
models and are summarized in table 2.
We compared model simulations with long-term (1970-1999) mean monthly discharge data from Dai *et al*
(2009).  We computed model total annual discharge (sum of surface and subsurface runoff) for the main
river basins in the permafrost region of North America (Mackenzie, Yukon) and Russia (Yenisei, Lena).
In particular, we compared (i) annual runoff anomalies, (ii) correlation coefficients and (iii) distributions
of annual discharge between gauge data and models' simulations for the 30-year period of 1970-1999.
Gauge stations from major permafrost river basins used for simulation comparison include (i) Arctic Red,
Canada ($67.46^0$N, $133.74^0$W) for Mackenzie River, (ii) Pilot Station, Alaska ($61.93^0$N $162.88^0$W) for
Yukon River, (iii) Igarka, Russia ($67.43^0$N, $86.48^0$E) for Yenisey River and (iv) Kusur, Russia ($70.68^0$N,
$127.39^0$E) for Lena River.

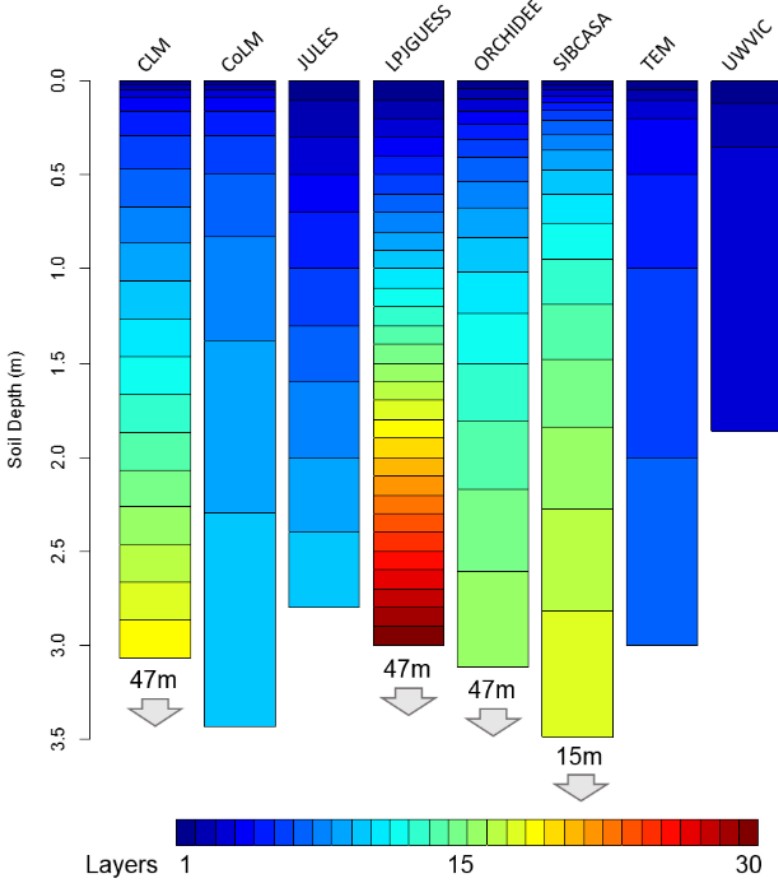

**Figure 1. Soil hydrologically-active column configuration for each participating model. Numbers**
**and arrows indicate full soil configuration of non-hydrologically active bedrock layers. Colors**
**represent the number of layers.**

**Table 1. Models description and driving datasets.**

| Model | Full Name | Climate Forcing Dataset | Model Reference | Short-Wave radiation[a] | Long-Wave Radiation[a] | Vapor Pressure[a] |
|---|---|---|---|---|---|---|
| CLM 4.5 | Community Land Model v4.5 | CRUNCEP4[b] | Oleson *et al* (2013) | Yes | Yes[c] | Yes |
| CoLM | Common Land Model | Princeton[d] | Dai *et al* (2003), Ji *et al* (2014) | Yes | Yes | Yes |
| JULES | Joint UK Land Environment Simulator model | WATCH (1901-2001)[e] | Best *et al* (2011) | Yes | Yes | Yes |
| ORCHIDEE-IPSL | Organising Carbon and Hydrology In Dynamic Ecosystems | WATCH (1901-1978)[e] | Gouttevin, I. *et al* (2012), Koven *et al* (2009), Krinner *et al* (2005) | Yes | Yes | Yes |

| | | | | | | |
|---|---|---|---|---|---|---|
| LPJGUESS | Lund-Postdam-Jena dynamic global veg model | CRU TS 3.1[f] | Gerten *et al* (2004), Wania *et al* (2009b, 2009a) | Yes | No | No |
| SiBCASA | Simple Biosphere/Carnegie-Ames-Standford Approach model | CRUNCEP4[b] | Schaefer *et al* (2011), Bonan (1996), Jafarov, E. and Schaefer (2016) | Yes | Yes | Yes |
| TEM604 | Terrestrial Ecosystem Model | CRUNCEP4[b] | Hayes *et al* (2014, 2011) | Yes | No | No |
| UW-VIC | Univ. of Washignton Variable Infiltration Capacity model | CRU[f], Udel[h] | Bohn *et al* (2013) | Internally calculated | Internally calculated | Yes |

[a]Simulations driven by temporal variability

[b]Viovy and Ciais (http://dods.extra.cea.fr/)

[c]Long-wave dataset not from CRUNCEPT4

[d]Sheffield *et al* (2006) (http://hydrology.princeton.edu/data.pgf.php)

[e]http://www.eu-watch.org/gfx_content/documents/README-WFDEI.pdf

[f]Harris *et al* (2014), University of East Anglia Climate Research Unit (2013)

[g]Mitchell and Jones (2005) for temperature

[h]Willmott and Matsuura (2001) for wind speed and precipitation with corrections (see Bohn et al. 2013).

**Table 2. Hydrology and soil thermal characteristics of participating models.**

| | Hydrology | | | | | | | Soil Thermal Properties | | | |
|---|---|---|---|---|---|---|---|---|---|---|---|
| Model | Evapotranspiration approach | Root water uptake | Infiltration | Water table | Soil Water Storage and Transmission | Groundwater Dynamics | Soil-ice impact | Snow | Soil thermal dynamics approach | Unfrozen Water effects on Phase Change | Moss insulation | Organic soil insulation |
| CLM 4.5 | Sum of canopy evaporation, transpiration, and soil evaporation | Macroscopic approach | Saturation-excess runoff $F_{sat}$=f($z_{wt}$) | Niu et al. (2007); perched water table possible if ice layer present | Richard's equation (Clapp Hornberger functions) | Base flow from TOPMODEL concepts, unconfined aquifer (Niu et al. 2007) | Impacts hydrologic properties through power-law ice impedance (Swenson et al., 2012) | Multi-layer dynamic (5 max) | Multi-layer Finite Difference Heat Diffusion | Yes | No | Yes |
| CoLM | BATS and Philip's (1957) | Macroscopic approach | Saturation-excess runoff $F_{sat}$=f($z_{wt}$) | Simple TOPMODEL | Richard's equation (Clapp Hornberger functions) | Base flow from TOPMODEL | Impacts hydrologic properties through power-law ice impedance | Multi-layer dynamic (5 max) | Multi-layer Finite Difference Heat Diffusion | No | No | No |
| JULES | Sum of ET, soil evaporation and moisture storages (e.g. lakes, urban) minus surface resistance | Macroscopic approach | Saturation-excess runoff $F_{sat}$=f($z_{wt}$) or $F_{sat}$=f(θ) | TOPMODEL or Probability Distribution Model | Richard's equation (Clapp Hornberger/van Genuchten functions) | Base flow from TOPMODEL | Hydraulic conductivity and suction determined by unfrozen water content (Brooks and Corey functions) | Multi-layer dynamic (3 max) | Multi-layer Finite Difference Heat Diffusion | Yes | No | No |
| ORCHIDEE-IPSL | Sum of bare soil, interception loss and plant transpiration for different veg PFTs in grid cell. | Macroscopic approach, water uptake different among cell veg PFTs (de Rosnay and Polcher, 1998) | Saturation-excess runoff $F_{sat}$=f(θ) | TOPMODEL | Richard's equation (van Genuchten functions) | None | "Drying=Freezing" approximation (Gouttevin et al 2012) | Multi-layer dynamic (7 max) | 1D Fourier Solution | Yes | No | Yes |
| LPJ-GUESS | Sum of Interception loss, plant transpiration and evaporation from soil. Gerten et al (2004) | Fractional water uptake from different soil layers according to prescribed root distribution. (Wania et al., 2009a,b) | Depends on soil moisture and layer thickness. Declines exponentially with soil moisture | Uniform, and only for wetland grid cell (Wania et al., 2009a,b) | Analog to Darcy's Law, percolation rate depends on soil texture conductivity and soil wetness (Haxeline and Prentice, 1996). | Base flow is based on the exponential function to estimate percolation rate | Impacts hydrologic properties through power-law ice impedance | Multi-layer dynamic (3 max) | Multi-layer Finite Difference Heat Diffusion | No | No | No |
| SiBCASA | Sum of ground evaporation, surface dew, canopy ET and canopy dew (Bonan, 1996) | Macroscopic approach | Infiltration approach in non-saturated porous media described by Darcy's law | Niu et al. (2007); perched water table possible if ice layer present | Richard's equation (Clapp Hornberger functions) | Base flow from TOPMODEL concepts, unconfined aquifer (Niu et al. 2007) | Impacts hydrologic properties through power-law ice impedance | Multi-layer dynamic (5 max) | Multi-layer Finite Difference Heat Diffusion | Yes | No | Yes |
| TEM-604 | Jenson-Haise potential ET (PET, Jenson and Haise 1963). Actual ET is calculated based on PET, water availability and leaf mass. | Based on the proportion of actual ET to potential ET | Field capacity-excess runoff (Thornthwaite and Mather 1957) | none | one-layer bucket | none | none | Multi-layer dynamic (9 max) | Multi-layer Finite Difference Heat Diffusion | No | Yes | No |
| UW-VIC | Sum of canopy interception, veg. transpiration and soil evaporation (Liang et al. 1994) | Based on reference ET and soil wilting point | Saturation-excess runoff $F_{sat}$=f(θ) | Microtopography | From infiltration rate and infiltration shape parameter (Liang et al. 1994). No lateral flow between model grids | Base flow from Arno model conceptualization (Francini and Pacciani 1991) | Impacts hydrologic properties through power-law ice impedance | Bulk-layer dynamic (2 max) | Multi-layer Finite Difference Solution | Yes | No | Yes |

**2. Results**
**3.1 Soil Moisture**

Air temperature forcing from greenhouse-gas emissions shows an increase of ~15°C in the permafrost
domain over the simulation period (Figure 2a). With increases in air temperature, models project an
ensemble mean decrease of ~13 million km$^2$ (91%) of the permafrost domain by 2299 (Figure 2b).
Coincident with these changes, most models projected a long-term drying of the near-surface soils when
averaged over the permafrost landscape (Figure 2c). However, the simulations diverged greatly with
respect to both the permafrost-domain average soil moisture response and their associated spatial patterns
(Figure 2c, 3). The models' ensemble mean indicated a change of -10% in near-surface soil moisture for
the permafrost region by year 2299, but the spread across models was large.  COLM and LPJGUESS
simulate an increase in soil moisture of 10% and 48%, respectively. CLM, JULES, TEM6 and UWVIC
exhibit qualitatively similar decreasing trends in soil moisture ranging between -5% and -20%. SIBCASA
and ORCHIDEE projected a large soil moisture change of approximately -50% by 2299. Spatially,
models show diverse wetting and drying patterns and magnitudes across the permafrost zone (Figure 3).
Several models tend to get wetter in the colder northern permafrost zones and are more susceptible to
drying along the southern permafrost margin.  Other models, such as TEM6 and UWVIC show the
opposite pattern with drying more common in the northern part of the permafrost domain.

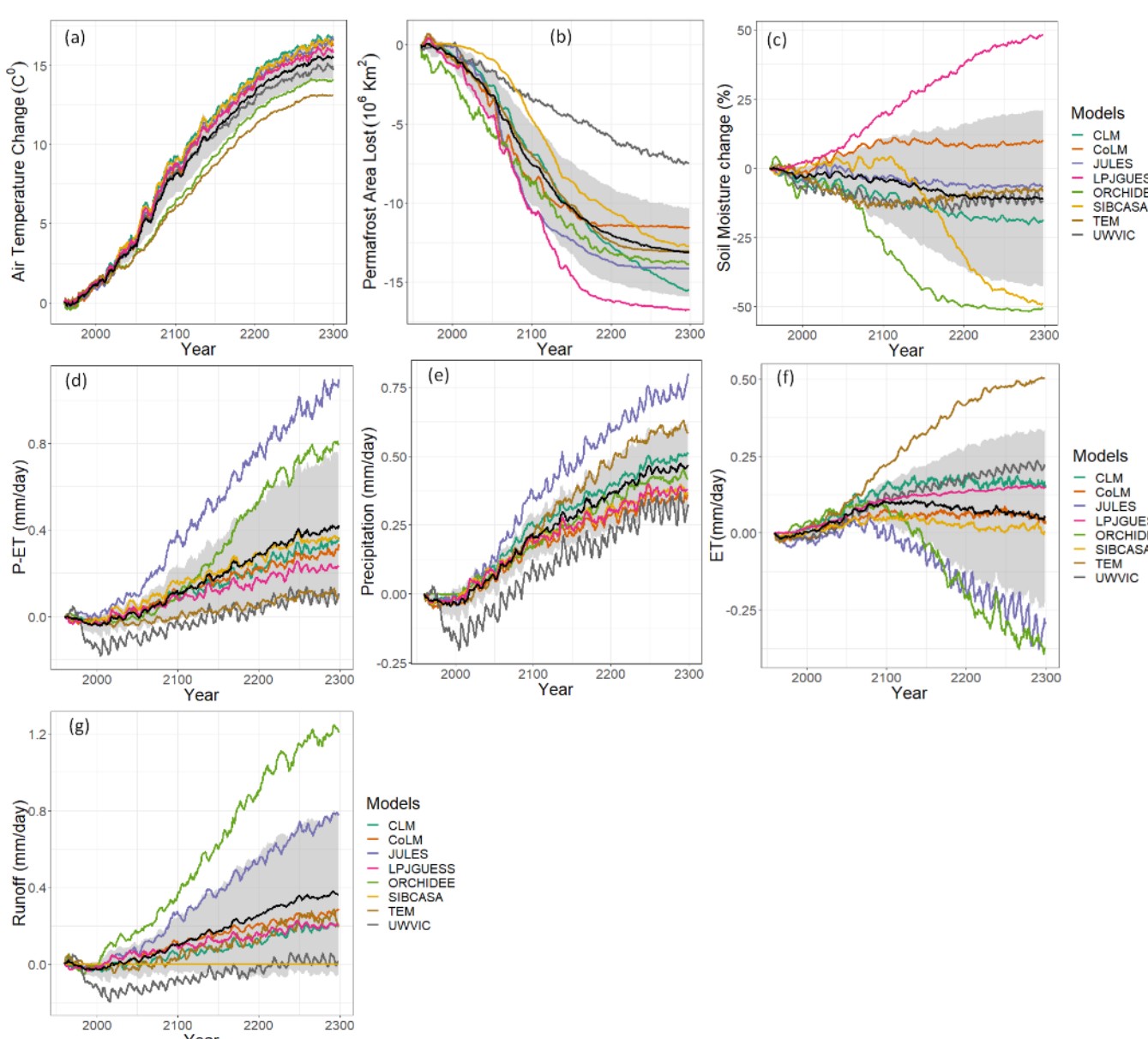

**Figure 2. Simulated annual mean changes in air temperature, near-surface permafrost area, near-**
**surface soil moisture and hydrology variables relative to 1960 (RCP 8.5). Annual mean is computed**
**from monthly output values. The black line represents the models' ensemble mean and the gray**
**area is the ensemble standard deviation. Figures d, e, f, and g are represented as change from 1960**
**values. Time series are smoothed with a 7-year running mean for clarity and calculated over the**
**initial permafrost domain of each model in 1960 for latitude >45⁰N.**

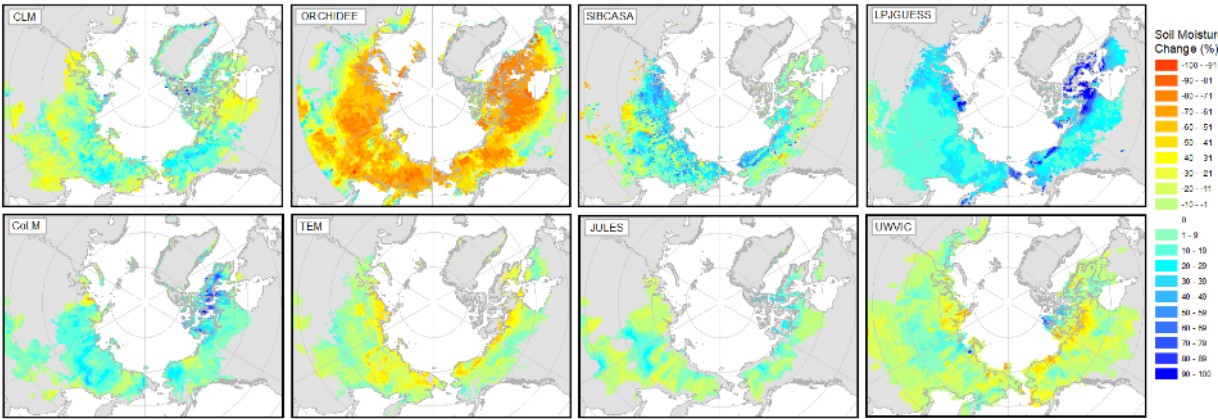

**Figure 3. Spatial variability of projected changes in surface soil moisture (%) among models.**
**Depicted changes are calculated as the difference between the 2071 to 2100 average and the 1960 to**
**1989 average. Colored area represents the initial simulated permafrost domain of 1960 for each**
**model.**

**3.2 Drivers of Soil Moisture Change**

To understand why models projected upper soil drying despite increases in the net precipitation (P-ET)
into the soil, we examined whether or not increases in active layer thickness (ALT) and/or complete thaw
of near-surface permafrost could be related to surface soil drying of the top 0-20cm ALT. We observed a
general significant negative correlation in most models (except SIBCASA, LPJGUESS) where cells with
greater increases in active layer thickness have greater drying (decrease) in near-surface soil moisture
(Figure 4). However, there is a large spread between soil moisture and ALT changes (Figure 4). This
spread may be influenced by many interacting factors that can be difficult to assess directly and are out of
the scope of this study. In addition, the coarse soil column discretization in UWVIC limited this analysis
for this model (Figure 1). However, most models show some indication that as the active layer deepens,
soils tend to get drier at the surface.

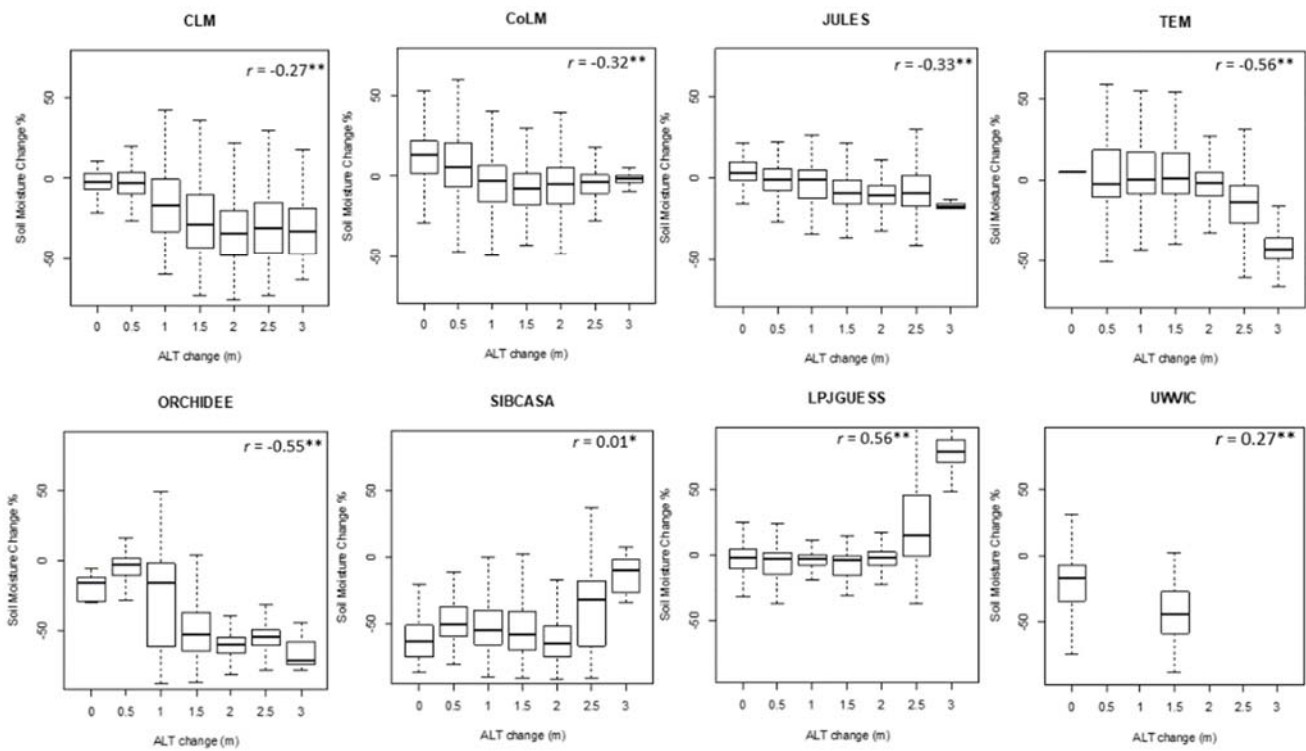

**Figure 4. Responses of August near-surface (0-20cm) soil moisture to ALT changes. Each box**
**represents a range of ±0.25m of ALT change. ALT and soil moisture change are calculated as the**
**2290-2299 average minus the 1960-1989 average for cells in the initial permafrost domain of 1960.**
**For cells where ALT exceeded 3 meters (no permafrost) during 2270-2299 period, we subtracted**
**the initial active layer thickness (1960-1989 average) to 3 meters. Population Pearson correlations**
**(*r*) significant at *p<0.01 and **p<2e-16.**
**3.3 Precipitation, ET, and Runoff**
Models may project surface soil drying but the hydrological pathways through which this drying occurs
appears to differ across models. The diversity of precipitation partitioning (Figure 5) demonstrates that
specific representations and parameterizations for ET and runoff are not consistent across models. Though
some models maintain a similar R/P ratio throughout the simulation (e.g., CLM, COLM, LPJGUESS),
others show shifts from an ET-dominated system to a runoff-dominated system (e.g. JULES) and vice
versa (e.g. TEM6 and UWVIC).
Evapotranspiration from the permafrost area is projected to rise in all models driven by warmer air
temperatures and more productive vegetation, but the amplitude of that trend varies widely. The average
projected evapotranspiration increase is 0.1±0.1mm/day (mean ± SD, hereafter) by 2100, which
represents about a 25% increase over 20[th] century levels. Beyond 2100, the ET projections diverge
(Figure 2e).
Runoff is also projected to increase with projections across models being highly variable (Figure 2g). The
change in the models' ensemble mean between 1960-2299 was 0.2±0.2 mm/day. CLM, COLM,
LPJGUESS and TEM6 simulated runoff changes of 0.2 to 0.3 mm/day by 2299. UWVIC exhibit small to
null changes in runoff while SIBCASA shows surface runoff only.
Comparison between gauge station data and runoff simulations from the major river basins in the
permafrost region shows that most models agree on the long term timing (Figure 6, Table 3) but the
magnitude is generally underestimated (Figure 7). The gauge discharge mean for the four river basins is
$219 \pm 36$ mm/yr compared to the models' ensemble mean of $101 \pm 82$ mm/yr for the period 1970-1999.
Excluding SIBCASA, the models' ensemble mean is $134 \pm 69$ mm/yr. However, models show reasonable
correlations between runoff output and observed annual discharge time series (Table 3). SIBCASA
horizontal subsurface runoff was disabled on the simulation because it tended to drain the active layer
completely, resulting in very low and unrealistic soil moisture. Therefore, SIBCASA runoff values shown
in this study are only for surface runoff.
The net water balance (P-ET-R) is projected to increase for most models with precipitation increases
outpacing the sum of ET and runoff changes. All models except TEM6 show an increase in the net water
balance over the simulation period which suggests that models are collecting soil water deeper in the soil
column, presumably in response to increasing ALT, even while the top soil layers dry.

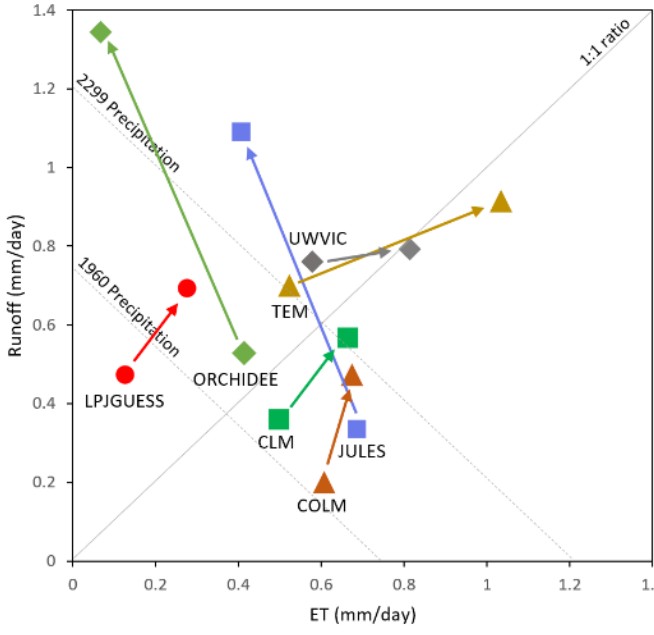

**Figure 5. Precipitation partitioning between total runoff and evapotranspiration for participating**
**models. Markers and arrows indicate the change from initial period (1960-1989 average) to final**
**period (2270-2299 average). Diagonal dashed lines represent the ensemble rainfall mean for the**
**initial (0.74 mm/day) and final (1.2 mm/day) simulation years. At any point along the dashed**
**diagonals, runoff and ET sum to precipitation.**

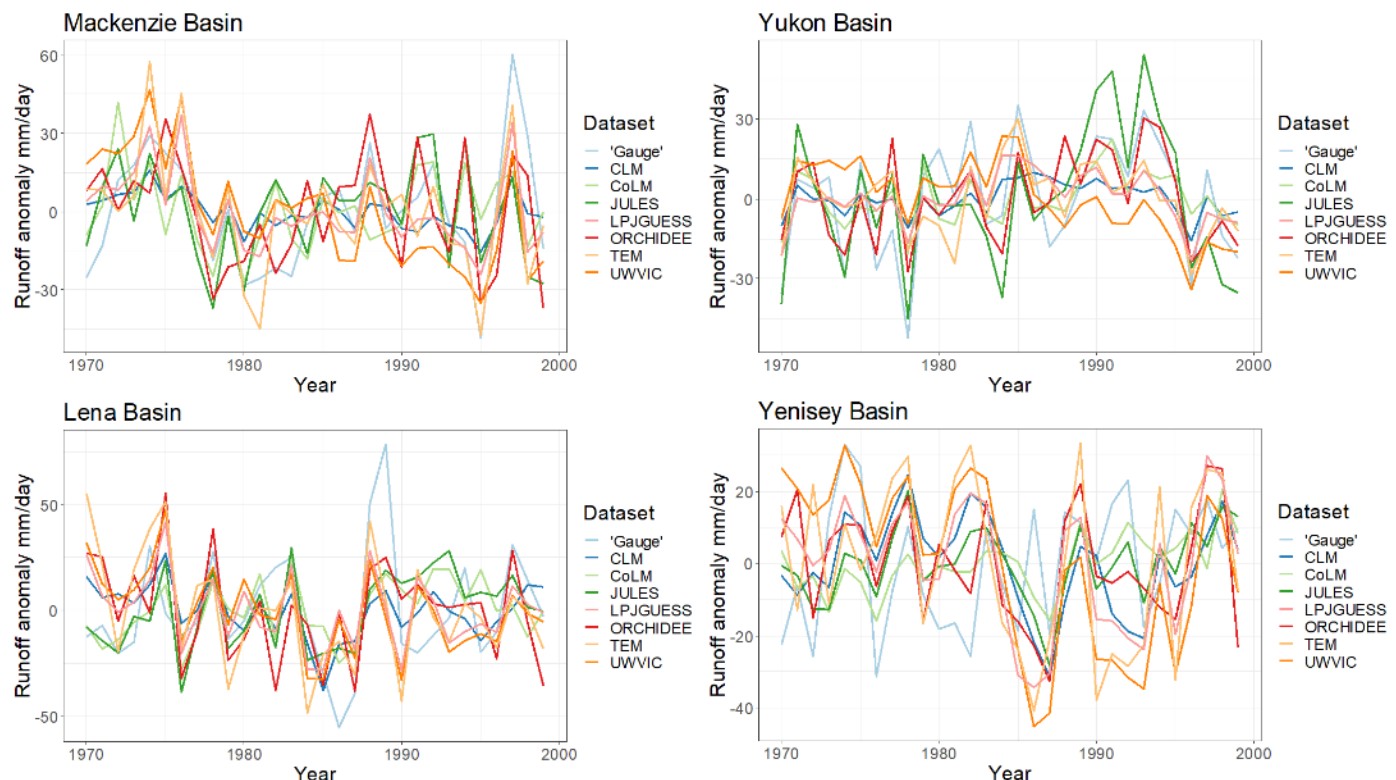


**Figure 6. Runoff anomaly comparison between gauge data and models simulations for the period 1970-1999.**


**Table 3. Correlation coefficients between simulated annual total runoff and gauge mean annual discharge 1970 to 1999. SIBCASA correlations are for surface runoff.**

| | River Basin | | | | |
|---|---|---|---|---|---|
| Model | Mackenzie | Yukon | Yenisey | Lena | Avg. |
| CLM | 0.70 | 0.64 | 0.08 | 0.46 | 0.47 |
| ORCHIDEE | 0.57 | 0.69 | 0.36 | 0.37 | 0.50 |
| LPJGGUESS | 0.68 | 0.71 | 0.14 | 0.35 | 0.47 |
| TEM | 0.66 | 0.56 | 0.16 | 0.40 | 0.45 |
| SIBCASA | 0.49 | 0.21 | 0.08 | 0.29 | 0.27 |
| JULES | 0.41 | 0.77 | 0.34 | 0.51 | 0.51 |
| COLM | 0.38 | 0.76 | 0.27 | 0.46 | 0.47 |
| UWVIC | 0.44 | 0.38 | 0.02 | 0.31 | 0.29 |
| Avg. | 0.54 | 0.59 | 0.18 | 0.40 | |


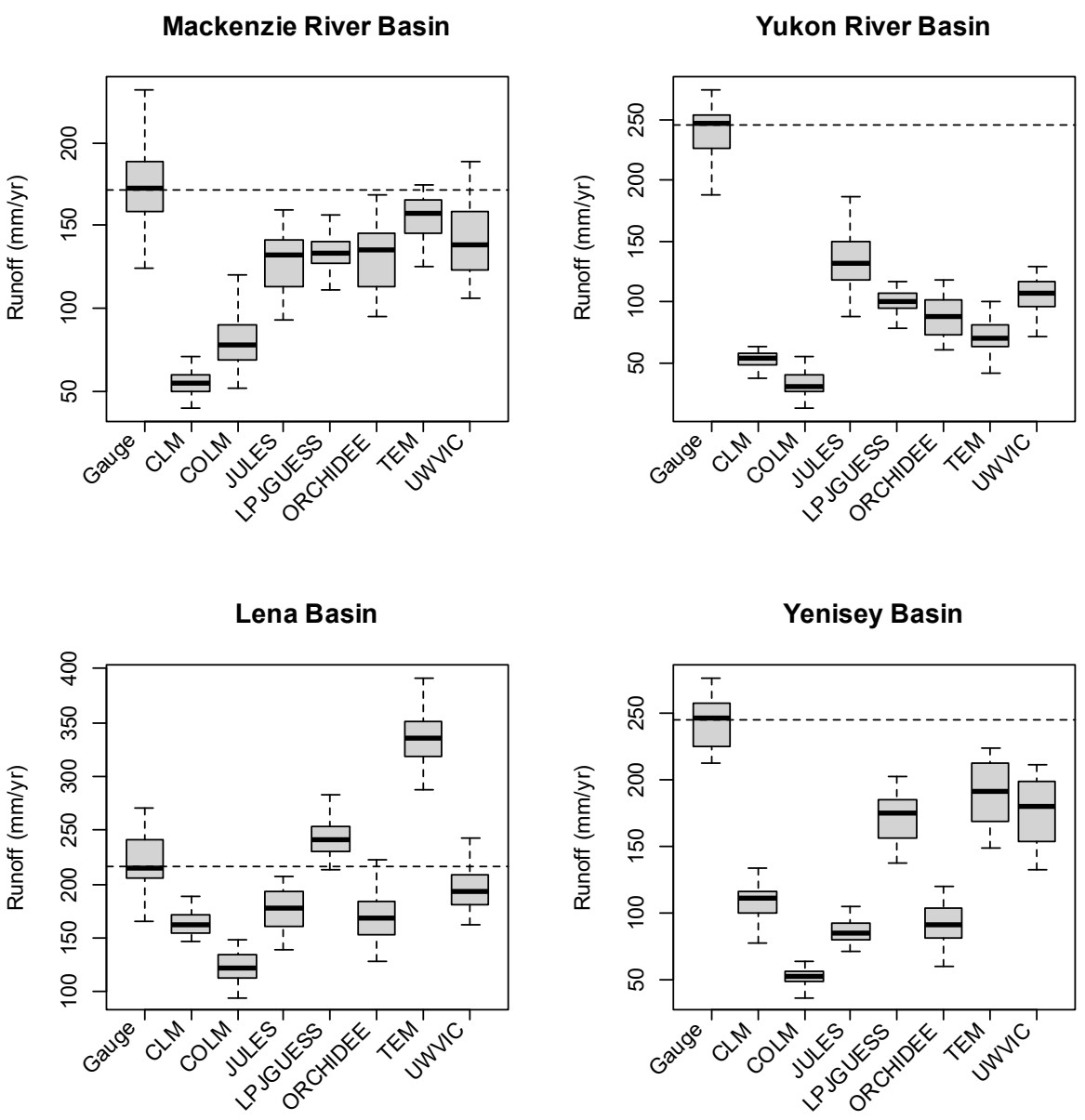

Figure 7. Discharge comparison between gauge station data and model output for each river basin.
Dashed line indicates mean annual discharge at gauge station. Boxplots derived from mean annual
discharge (total runoff) simulations for the period of 1970 to 1999.

## 4. Discussion

This study assessed near-surface soil moisture and hydrology projections in the permafrost region using
widely-used land models that represent permafrost. Most models showed near-surface drying despite the
externally-forced intensification of the water cycle driven by climate change. Drying was generally
associated with increases of active layer thickness and permafrost degradation in a warming climate. We
show that the timing and magnitude of projected soil moisture changes vary widely across models,

pointing to an uncertain future in permafrost hydrology and associated climatic feedbacks. In this section,
we review the role of projected permafrost loss and active layer thickening on soil moisture changes and
some potential sources of variability among models. In addition, we comment on the potential effects of
soil moisture projections on the permafrost carbon-climate feedback. It is important to note that this study
is more qualitative in nature and does not focus on the detail of magnitude or spatial patterns of model
signatures.
**4.1 Permafrost degradation and drying**
Increases in net precipitation and the counterintuitive drying of the top soil in the permafrost region
suggests that soil column processes such as changes in active layer thickness (ALT) and activation of
subsurface drainage with permafrost thaw are acting to dry the top soil layers (Figure 8a). In general,
models represent impermeable soils when frozen. Then, as soils thaw at progressively depths in the
summer, liquid water infiltrates further into the active layer draining deeper into the thawed soil column
(Avis et al., 2011; Lawrence et al., 2015; Swenson et al., 2012). However, relevant soil column processes
related to thermokarst by thawing of excess ground ice (Lee et al., 2014) are limited in these simulations
despite their significant occurrence in the permafrost region (Olefeldt et al., 2016). As permafrost thaws,
ground ice melts, potentially reducing the volume of the soil column and changing the hydrological
properties of the soil (Aas et al., 2019; Nitzbon et al., 2019). This would occur where soil surface
elevation drops through sudden collapse or slow deformation by an amount equal to or greater than the
increased depth of annual thaw (Figure 8b). This mechanism, not represented in current large-scale
models, could result in projected increases or no change in the water table over time as observed by long-
term studies (Andresen and Lougheed, 2015; Mauritz et al., 2017; Natali et al., 2015). Subsidence of 12-
13 cm has been observed in Northern Alaska over a five year period, which represents a volume loss of
about 25% of the average ALT for that region (~50cm) (Streletskiy et al., 2008). These lines of evidence
may suggest that permafrost thaw may not dry the Arctic as fast as simulated by land models but rather
maintain or enhanced soil water saturation depending on the water balance of the modeled cell column.

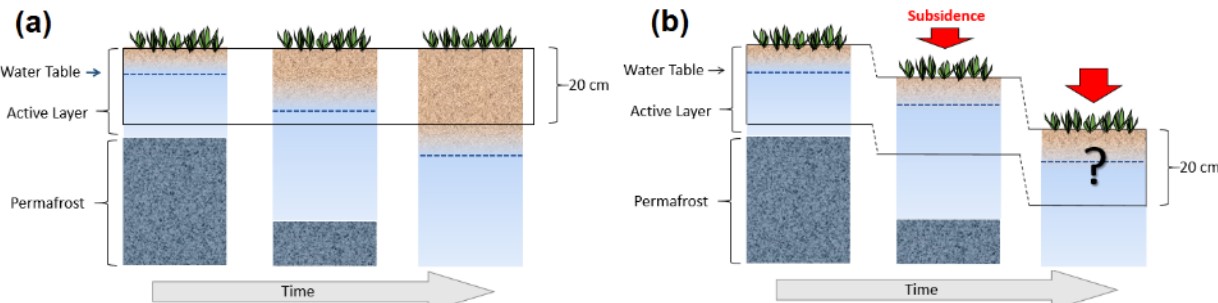

**Figure 8. Schematic of changes in the soil column moisture (a) without subsidence (current models)**
**and (b) with subsidence from thawing ice-rich permafrost (not represented by models), a process**
**that may accumulate soil moisture and slow down drying over time.**
Recent efforts have been made to address the high sub-grid heterogeneity of fine-scale mechanisms
including soil subsidence (Aas et al., 2019), hillslope hydrology, talik and thermokarst development
(Jafarov et al., 2018), ice wedge degradation (Abolt et al., 2018; Liljedahl et al., 2016; Nitzbon et al.,
2019), vertical and lateral heat transfer on permafrost thaw and groundwater flow (Kurylyk et al., 2016)
and lateral water fluxes (Nitzbon et al., 2019). These processes are known to have a major role on surface
and subsurface hydrology and their implementation in large scale models is needed. Other important
challenges in land models' hydrology include representation of the significant area dynamics of the
ubiquitous smaller, shallow water bodies observed over recent decades (Andresen and Lougheed, 2015;
Jones et al., 2011; Roach et al., 2011; Smith et al., 2005). These systems are either lacking in simulations
(polygon ponds and small lakes) or assumed to be static systems in simulations (large lakes). The
implementation of surface hydrology dynamics and permafrost processes in large-scale land models will
help reduce uncertainty in our ability to predict the future hydrological state of the Arctic and the
associated climatic feedbacks. It is important to note that all these processes require data for model
calibration, verification and evaluation, that is commonly absent at large scales. Permafrost hydrology
will only advance through synergistic efforts between field researchers and modelers.
**4.2 Uncertainty in soil moisture and hydrology simulations**
Differences in representations of soil thermal dynamics can directly affect hydrology through timing of
the freezing-thawing cycle and by altering the rates of permafrost loss and subsurface drainage (Finney et
al., 2012). McGuire et al. (2016) and Peng et al. (2016) show that these models exhibit considerable
differences in permafrost quantities such as active layer thickness, and the mean and trends in near-
surface (0-3m) permafrost extent, even though all the models are forced with observed climatology.
However, these differences are smaller than those seen across the CMIP5 models (Koven et al., 2013). All
models except ORCHIDEE employ a multi-layer finite difference heat diffusion for soil thermal
dynamics (Table 2). Organic soil insulation, snow insulation, and unfrozen water effects on phase change
are the most common structural differences among models for soil thermal dynamics but do not explain
the variability in the simulated changes in ALT and permafrost area as shown by McGuire *et al* (2016).
Half of the participating models include organic matter in the soil properties (CLM, ORCHIDEE,
SIBCASA, UWVIC) which can significantly impact soil thermal properties and lead to an increase in the
hydraulic conductivity of the soil column, thereby enhancing drainage and redistribution of water in the
soil column. Soil vertical characterization is another important aspect for soil thermal dynamics and
hydrology (Chadburn et al., 2015; Nicolsky et al., 2007). Lawrence et al (2008) indicated that a high-
resolution soil column representation is necessary for accurate simulation of long term trends in active
layer depth. However, McGuire *et al* (2016) showed that soil column depth did not clearly explain
variability of the simulated loss of permafrost area across models.
Water table representation can result in a first order effect on soil moisture. Most models (CLM, COLM,
SIBCASA and ORCHIDEE) use some version of TOPMODEL (Niu et al., 2007), which employs a
prognostic water table where sub-grid scale topography is the main driver of soil moisture variability in
the cell. However, water table is not explicitly represented in other models such as LPJGUESS, which has
a uniform water table which is only applied for wetland areas. In addition to water table, storage and
transmission of water in soils is a fundamental component of an accurate representation of soil moisture
(Niu and Yang, 2006). The representation of soil water storage and transmission varies across models
from Richards equations based on Clapp Hornberger and/or van Genuchten (1980) functions (e.g CLM,
CoLM, SIBCASA, ORCHIDEE) to a simplified one layer bucket (e.g. TEM6).  It is also important to
note that most models differ in their numerical implementations of processes, such as water movement
through frozen soils (Gouttevin, I. et al., 2012; Swenson et al., 2012), and in the use of iterative solutions
and vertical discretization of water transmission (De Rosnay et al., 2000).
Differences in representation of vertical fluxes through evapotranspiration (ET) are also likely adding to
the high variability in soil moisture projections. ET sources (e.g. interception loss, plant transpiration, soil
evaporation) were similar across models but had different formulations (Table 2). The diversity of ET
implementations (e.g. evaporative resistances from fractional areas, etc.) and of vegetation maps used by
the modelling groups (Ottlé et al., 2013) can also contribute to the big spread on the temporal simulations
for ET and soil moisture. Along with projected increases in ET, net precipitation (P-ET) is projected to
increase for all models suggesting that drying is not attributed only to soil evaporation, and the increasing
net water balance (P-ET-R) proposes that models are storing water deeper in the soil column as
permafrost near the surface thaws.
Despite runoff improvements (Swenson et al., 2012), underestimation of river discharge has been a
challenge in previous versions in models (Slater et al., 2007). The differences between models and
observations in mean annual discharge may stem from several sources. Particularly, the substantial
variation in the precipitation forcing for these models (Figure 2e). This is attributed, in part, to the sparse
observational networks in high latitudes. River discharge at high latitudes can differ substantially when
different reanalysis forcing datasets are used. For example, river discharge for Arctic rivers differs
substantially in CLM4.5 simulations when forced with GSWP3v1 compared to CRUNCEPv7 reanalysis
datasets (not shown, runoff for MacKenzie, +32%; Yukon, +78%; Lena, -2%; Yenisey, +22%). Other
factors include potential deficiencies in the parameterization and/or implementation of ET and runoff
processes as well as vegetation processes.

**4.3 Implications for the permafrost carbon-climate feedback**

If drying of the permafrost region occurs, carbon losses from the soil will be dominated by $CO_2$ as a result
of increased heterotrophic respiration rates compared to moist conditions (Elberling et al., 2013;
Oberbauer et al., 2007; Schädel et al., 2016). With projected drying, $CH_4$ flux emissions will slow down
by the reduction of soil saturation and inundated areas through lowering the water table in grid cells
(Figure 8A). In a sensitivity study using CLM, the slower increase of methane emissions associated with
surface drying could potentially lead to a reduction in the Global Warming Potential of permafrost carbon
emissions by up to 50% compared to saturated soils (Lawrence et al., 2015). However, we need to also
consider that current land models lack representation of important $CH_4$ sources and pathways in the
permafrost region such as lake and wetland dynamics that can counteract the suppression of $CH_4$ fluxes
by projected drying. Seasonal wetland area variation, which is not represented or is poorly represented in
current models, can contribute to a third of the annual $CH_4$ flux in boreal wetlands (Ringeval et al., 2012).
Although this manuscript may raise more questions than answers, this study highlights the importance of
advancing hydrology and hydrological heterogeneity in land models to help determine the spatial
variability, timing, and reasons for changes in hydrology of terrestrial landscapes of the Arctic. These
improvements may constrain projections of land-atmosphere carbon exchange and reduce uncertainty on
the timing and intensity of the permafrost carbon feedback.

**Data availability**

The simulation data analyzed in this manuscript is available through the National Snow and Ice Data
Center (NSIDC; http://nsidc.org). Inquires please contact Kevin Schaefer (kevin.schaefer@nsidc.org).

**Author contributions**

This manuscript is a collective effort of the modeling groups of the Permafrost Carbon Network (http://www.permafrostcarbon.org). C.G.A, D.M.L., C.J.W., A.D.M. wrote the initial draft with additional contributions of all authors. Figures prepared by C.G.A.

**Acknowledgements**

This manuscript is dedicated to the memory of Andrew G. Slater (1971 -2016) for his scientific contributions in advancing Arctic hydrology modeling. This work was performed under the Next-Generation Ecosystem Experiments (NGEE Arctic, DOE ERKP757) project supported by the Office of Biological and Environmental Research in the U.S. Department of Energy, Office of Science. The study was also supported by the National Science Foundation through the Research Coordination Network (RCN) program and through the Study of Environmental Arctic Change (SEARCH) program in support of the Permafrost Carbon Network. We also acknowledge the joint DECC/Defra Met Office Hadley Centre Climate Programme (GA01101) and the European Union FP7-ENVIRONMENT project PAGE21.

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
