# Peer review of "Soil Moisture and Hydrology Projections of the Permafrost 1"

_The Cryosphere, 2019_

## Referee Comment (RC1) · Anonymous Referee #1 · 12 Aug 2019

In this study, the authors compare eight land surface models in the PCN-MIP for their performance in modeling hydrological processes in the permafrost region. Authors examine the long-term change of surface (0-20 cm) soil moisture, finding a drying trend in most models. Authors attribute the drying trend to moisture infiltration as the active layer deepens. In modeling runoff, models tend to have structural limitations that underestimate runoff volume compared to observations. Authors conclude that this generation of land surface models

This is overall a necessary and useful study in terms of model intercomparison, presenting the capability of the current generation of land surface models in projecting the

hydrological state of the Arctic, which also gives some insights to the application of models. According to the cover letter, the authors have tuned the scope of this study to a model intercomparison along with many other revisions after the previous rejection. I think, however, some major issues still remain unclear or unaddressed in this manuscript, I, therefore, recommend a major revision.

Major issues:

1. Definition of permafrost in land models

In figure 1, it is unclear that if all soil layers showing here are hydrologically-active or not. I think authors here show all soil layers since for some models the soil layers are as deep as 47 meters, while in the figure caption authors call the figure "soil hydrological column configuration". As bedrock layers do not involve in hydrological processes, authors should make clear in the figure how many layers for each model are hydrologically active. More importantly, this unclear statement raises a question in the definition of permafrost in this study. In section 2.2 (line 122), the authors define permafrost grid points with ALT less than 3 meters. However, for some models (JULES, TEM, and UWVIC) showing in Figure 1, the deepest soil layer is less than 3 meters deep. Then how permafrost is defined in these three models? Furthermore, comparisons are somewhat unreasonable because of the way authors define permafrost regions in these 8 models. Showing in Figure 3, the permafrost region actually differs substantially from models. ORCHIDEE has probably the biggest permafrost area globally while JULES has the smallest one. Different regions could correspond to different climate zones and climate changes associated with global warming. At least some differences showing in Figure 2 and 4 are originated from such different permafrost regions. Comparison over the overlapped regions with permafrost for all 8 models could be a more reasonable approach.

2. Runoff in SIBCASA and other models

Figure 6 & 7 show that the annual mean runoff in SIBCASA for the period of 1970-1999

is close to zero with little-to-none inter-annual variability, which is of course fairly biased from gauge station data. But in Table 3, there is also some (although not high) degree of correlation between observation and SIBCASA-simulated runoff. For the Mackenzie Basin, it is not even the lowest. The runoff of SIBCASA is more "flawed" than "low" to me. Authors should give an explanation/speculation of why SIBCASA simulates such abnormal runoff. Is there any systematic error or technical failure? Or the model itself does not involve runoff modeling? If there is systematic error in SIBCASA-simulated runoff, authors should exclude it from correlation coefficient analysis.

Another potential deficiency of this model-observation comparison is the inconsistency of forcing data. Previous studies, which have also mentioned by authors in the discussion section, have suggested that different forcing data, even for reanalysis datasets that are observational-restricted, can cause some substantial biases in modeled variables. Since runoff is largely dependent on precipitation that is directly from the forcing, some difference of inter-annual variabilities of runoff and their difference between gauge data should be attributed to the difference in precipitation forcing.

3. Discussion in the uncertainty of soil and hydrology simulations

In the cover letter, authors mentioned that one of the reviewers in the previous submission rejected the manuscript partially because "The manuscript does not provide anything we don't already know from the literature, i.e. that the model results vary depending on what model and forcing you use". The authors tried to fix this issue by re-scoping the study and add discussions on the uncertainty of soil moisture and hydrology simulations. In my opinion, however, authors should work more to improve this part, showing how these differences contribute to the differed performances for different models in the result section.

Readers can actually expect all uncertainties authors discussed solely from Table 1 & 2, where different numerical implementations are listed. Of course, models could differ substantially that may cause differences in model output. In section 4.2, authors should

work more on linking the discussed uncertainty to the intercomparison results. For example, in line 317, the authors mentioned that involving organic matter could enhance drainage and redistribution of water in the soil column. Is there any evidence showing in the model intercomparison? Are models with organic matter involved showing greater drainage? And if not, is the signal covered up by some other more dominating physical processes? Similar discussion/comparison should be addressed as much as possible for all factors the authors mentioned in section 4.2. Otherwise, this part looks more like a literature review than discussion.

Minor issues:

Figure 2: Why the precipitation for UWVIC behaves abnormally in the historical period, which decreases substantially between 1970 and 2000? If without such decrease, the precipitation, P-ET, and runoff for UWVIC would possibly be fairly close to the average. As precipitation data is directly from the forcing data, authors should explain/discuss how the different forcing datasets in the historical period could bring biases to permafrost thermodynamics and hydrology, and if the biases in the historical period influence the simulation in the projected period.

Figure 3: Specifically for JULES, some Arctic coastal regions in Eurasia and Alaska are not defined as permafrost? Is it due to a lack of spatial resolution?

Figure 4: The Y-axis ticks for ORCHIDEE should be changed to the same as other sub-figures.

Table 3: P-values or significance tests should be addressed for these correlation coefficients.

Discussion section: In my opinion, section 4.1 and 4.2 should switch. As section 4.2 is more closely related and more important to the intercomparison results. Section 4.1, on the other hand, discusses the feature most involved models have not supported yet.

---

## Referee Comment (RC2) · Anonymous Referee #2 · 16 Aug 2019

Review of manuscript tc-2019-144 "Soil Moisture and Hydrology Projections of the Permafrost Region: A Model Intercomparison"

Overall comments

This paper presents an analysis and comparison of eight model simulations of changes to hydrology and soil moisture under changing permafrost conditions. The question of how Arctic landscapes will change in hydrological terms is both important and largely unresolved, so this is an relevant topic where a model intercomparison is useful.

The paper is well written and easy to understand. I think figures are mostly appropriate although some adjustments could improve clarity. I have some remarks on a number

of science points where I think the paper could be improved. With some clarifications and improvements, I think the paper could be a more useful contribution.

Major points

Abstract. The last sentence is quite general and states things that are very well known already. Could the abstract instead finish with a more interesting statement pointing out specific knowledge gaps or recommended directions of research?

106 Although method specifics can (hopefully) be obtained in the cited papers, I'd like a few more details here, for clarity. For one thing, it's not clear when the break point between historical, model-specific climate forcing and the common forcing took place. Was this at 1960 or at 2006?

114-117 Along the same lines, for clarity here: On what timescale did the historical CCSM4 climate forcing repeat?

134 Just to be clear, specify what years of model simulations were used for the comparison with 1970-1999 observations. I assume this is also long-term but is it the exact same period, or some other length?

157 Here the authors refer to the "permafrost domain", but this is not clearly defined in methods. Please clarify in the methods sections whether the study domain is, for each model, all cells with near-surface permafrost above 45 degrees N, as suggested on lines 121-123, or something else.

168-171 I am a bit dubious as to whether these patterns hold over longer-term analysis. If this statement is supported by the comparison of 10-year averages shown in Figure 3, I am unconvinced. See comment on that below.

Figure 3. Here the authors use a ten-year period to illustrate long-term spatial changes. This is way too short as decadal variability is clearly substantial for some models (Figure 2c). This should be a 30-year period.

191-193 I think this statement is not supported enough by the data. Either there is a relationship or not, and it would be easier to determine the likelihood of that with a simple x-y plot of the data rather than these box plots. As the authors note, the UWVIC model is not useful at all for this question due to its resolution. But for the box plots shown, I think the SIBCASA model clearly shows no tendency for more drying with ALT increase, which is not acknowledged. The statement should be modified to moderate this claim somewhat. Also, I am wondering at the use of short time periods again here, and would prefer a 30-year period comparison.

222-223 According to the text, JULES exhibits the highest runoff increase with 0.8 mm/day, but Figure 2g shows ORCHIDEE runoff increasing by 1.2 mm/day. Which is correct?

Minor and language points

110 The degree symbol seems to have been replaced by a 0 (zero character), at least on my computer.

161 Add "long-term" or "for the period after 2100" or similar to clarify that it's only after 2100 that most models stay on the drying side for soil moisture – up till then, about half of the models are close to zero change or wetting. I guess this is implicit with the talking of 2299 in the preceding sentence but still, just to be clear.

303 Change "large-scales" to "large scales".

392 Change "Study" to "The study".

Fig 1. The figure seems to show depths to 3.5 m but the caption says 3 m.

Fig 2. The caption says "Figures d, e, f, and g are represented as relative change from 1960 values". I think "relative change" implies a normalization which is not done here, so I suggest dropping "relative" from the above sentence.

Fig 7. At least in the pdf on my computer, the tick labels on the horizontal axis are

misaligned and show up inside the plot instead of outside. Please check.

---

## Short Comment (SC1) · 26 Aug 2019

In this manuscript the authors describe how most of the models examined project soil drying despite increases in net precipitation. Drying is attributed to increased drainage via active layer deepening and/or permafrost loss. This research is important, as soil moisture is a key control on the fate of carbon losses from soils. In Rawlins et al. (2013) my coauthors and I pointed to a likelihood of future soil drying based on simulations with the Permafrost Water Balance Model (PWBM, formerly the 'Pan-Arctic Water Balance Model'). In that study we discussed model validation results and explored potential thermal and hydrological changes for a representative area encompassing the

Bonanza Creek Experimental Forest research site in Alaska. Hydrological cycle intensification was manifested in higher spring SWE in the future simulation relative to present day, consistent with other research which points to future significant increases in cold season precipitation in Arctic regions. The model simulations showed that much of the snowmelt will become river runoff as opposed to soil recharge. This is intuitively expected, as the landscape is often frozen, or the active layer is very shallow, when thaw occurs. In turn, infiltration limits can also be easily reached at that time of year. A deeper snowpack in late winter would contribute to soil warming, which might lead to higher soil evaporation rates during early summer. These changes captured by the model simulations also suggest that drying may occur in areas of discontinuous and/or sporadic permafrost. Implications of permafrost thaw on cold season river discharge, subsurface runoff, and other hydrological quantities across northern Alaska were investigated using the PWBM as described in a manuscript currently under review for publication in The Cryosphere (Rawlins et al., 2019).

Rawlins, M.A., Nicolsky, D.J., McDonald, K.C. and Romanovsky, V.E., 2013. Simulating soil freeze/thaw dynamics with an improved pan‐Arctic water balance model. Journal of Advances in Modeling Earth Systems, 5(4), pp.659-675.

Rawlins, M. A., Cai, L., Stuefer, S. L., and Nicolsky, D.: Changing Characteristics of Runoff and Freshwater Export From Watersheds Draining Northern Alaska, The Cryosphere Discuss., https://doi.org/10.5194/tc-2019-28, in review, 2019.

Michael A. Rawlins Associate Director, Climate System Research Center University of Massachusetts-Amherst

[Figure]

[Figure]

Fig. 1.

---

## Author Comment (AC1) · 26 Aug 2019

Dear Reviewer, I sincerely appreciate taking the time to review this paper and provide very helpful comments and suggestions that significantly improved the clarity, flow and message of the manuscript. I addressed every comment you had and responses are below. Tracked changes are in the supplement pdf file. FYI- modified figures in text will have both versions where top figures will be the old version and bottom figures will be the new, corrected version. On behalf of all authors, Thank you. Christian Andresen

Reviewer #1 Major issues: 1. Definition of permafrost in land models In figure 1, it is unclear that if all soil layers showing here are hydrologically-active or not. I think au-

thors here show all soil layers since for some models the soil layers areas deep as 47 meters, while in the figure caption authors call the figure "soil hydrological column configuration". As bedrock layers do not involve in hydrological processes, authors should make clear in the figure how many layers for each model are hydrologically active. More importantly, this unclear statement raises a question in the definition of permafrost in this study. In section 2.2 (line 122), the authors define permafrost grid points with ALT less than 3 meters. However, for some models (JULES, TEM, and UWVIC) showing in Figure 1, the deepest soil layer is less than 3 meters deep. Then how permafrost is defined in these three models? Furthermore, comparisons are somewhat unreasonable because of the way authors define permafrost regions in these 8 models. Showing in Figure 3, the permafrost region actually differs substantially from models. ORCHIDEE has probably the biggest permafrost area globally while JULES has the smallest one. Different regions could correspond to different climate zones and climate changes associated with global warming. At least some differences showing in Figure 2 and 4 are originated from such different permafrost regions. Comparison over the overlapped regions with permafrost for all 8 models could be a more reasonable approach.

Authors response: These are important points that needed clarification in the manuscript. Thank you for highlighting them. Changes: -We modified the footnote of Figure 1 to clarify the hydrology layers of the models: "Figure 1. Soil hydrological-active column configuration for each participating model. Numbers and arrows indicate full soil configuration of models of non-hydrologically active bedrock layers. Colors represent the number of layers. " -We also clarified the permafrost estimation for the top 3m soil column which is slightly different among models due to its soil configuration layers ranging from 2-3m. Line 123 now reads: "we define a grid cell as containing near-surface permafrost if the annual monthly maximum active layer thickness (ALT) is at or less than the 3m depth layer depending on the model soil configuration (Figure 1)" -Regarding differences in permafrost extents across models, we decided to compare the full permafrost extent for each model rather than a subset to be representative for each model. The temperature (forcing) differences from the models, and thus, different areas, are shown in figure 2a and did not raised main concerns. However, we added clarification and highlighted this in the first paragraph of methods section 2.2: L121-123 "This qualitative hydrology comparison was based on the full permafrost domain in each model rather than a common subset among models in order to fully portray the overall changes in permafrost hydrology for participating models."

2. Runoff in SIBCASA and other models Figure 6 & 7 show that the annual mean runoff in SIBCASA for the period of 1970-1999 is close to zero with little-to-none inter-annual variability, which is of course fairly biased from gauge station data. But in Table 3, there is also some (although not high) degree of correlation between observation and SIBCASA-simulated runoff. For the Mackenzie Basin, it is not even the lowest. The runoff of SIBCASA is more "flawed" than "low" to me. Authors should give an explanation/speculation of why SIBCASA simulates such abnormal runoff. Is there any systematic error or technical failure? Or the model itself does not involve runoff modeling? If there is systematic error in SIBCASA-simulated runoff, authors should exclude it from correlation coefficient analysis. Another potential deficiency of this model-observation comparison is the inconsistency of forcing data. Previous studies, which have also mentioned by authors in the discussion section, have suggested that different forcing data, even for reanalysis datasets that are observational-restricted, can cause some substantial biases in modeled variables. Since runoff is largely dependent on precipitation that is directly from the forcing, some difference of inter-annual variabilities of runoff and their difference between gauge data should be attributed to the difference in precipitation forcing.

Authors response: We clarified the issues of low runoff in SIBCASA in the results and excluded it from Figures 5, 6 & 7 to avoid confusion and make the paper clearer. We kept SIBCASA in the correlation coefficient table 3 but highlighted that the analysis was for surface runoff only Changes: -Table 3 header: "Correlation coefficients between simulated annual total runoff and gauge mean annual discharge 1970 to 1999. SIBCASA correlations are for surface runoff." -We also added the following explanatory statement to the results: L239- "SIBCASA horizontal subsurface runoff was disabled on the simulation because it tended to drain the active layer completely, resulting in very low and unrealistic soil moisture. Therefore, SIBCASA runoff values shown in this study are only for surface runoff."

3. Discussion in the uncertainty of soil and hydrology simulations. In the cover letter, authors mentioned that one of the reviewers in the previous sub-mission rejected the manuscript partially because "The manuscript does not provide anything we don't already know from the literature, i.e. that the model results vary depending on what model and forcing you use". The authors tried to fix this issue by re-scoping the study and add discussions on the uncertainty of soil moisture and hydrology simulations. In my opinion, however, authors should work more to improve this part, showing how these differences contribute to the differed performances for different models in the result section. Readers can actually expect all uncertainties authors discussed solely from Table 1 &2, where different numerical implementations are listed. Of course, models could differ substantially that may cause differences in model output. In section 4.2, authors should work more on linking the discussed uncertainty to the inter comparison results. For ex-ample, in line 317, the authors mentioned that involving organic matter could enhance drainage and redistribution of water in the soil column. Is there any evidence showing in the model intercomparison? Are models with organic matter involved showing greater drainage? And if not, is the signal covered up by some other more dominating physical processes? Similar discussion/comparison should be addressed as much as possible for all factors the authors mentioned in section 4.2. Otherwise, this part looks more like a literature review than discussion.

Authors response: These are certainly important points for this study. Particularly, linking the uncertainty to differences processes will be very helpful for the science community. It is important to note that this study is a qualitative analysis (i.e. wetting vs drying) and does not focus on the details of magnitude and spatial patterns of the models signatures. Nonetheless, the manuscript originally addressed some of the uncertainty sources (e.g. organic matter, runoff, etc) for each model with the help of the modeling groups as "potential" causes of performance. However, the first review of the manuscript was discontent with these speculations and the lack of evidence to support them. Pinpointing these processes directly was difficult and required additional simulations. Therefore, we removed these from the manuscript and we focused on the main modelling challenges (e.g. ALT, soil thermal dynamics, ET, etc.) and supported the statements with literature. Changes: To clarify and remind the reader the focus of the paper, we added the following sentence in the first paragraph of discussion L279-281: "It is important to note that this study is more qualitative in nature and does not focus on the detail of magnitude or spatial patterns of model signatures." Minor issues:

Figure 2: Why the precipitation for UWVIC behaves abnormally in the historical period, which decreases substantially between 1970 and 2000? As precipitation data is directly from the forcing data, authors should explain/discuss how the different forcing datasets in the historical period could bring biases to permafrost thermodynamics and hydrology, and if the biases in the historical period influence the simulation in the projected period. Authors response: Discussion of how different forcing datasets influence projections it is certainly an important topic. However, in this manuscript we only focused on the overall trend of drying or wetting in these models rather than focusing in the detail at the magnitude and/or spatial patterns of the model signatures. No changes made

Figure 3: Specifically, for JULES, some Arctic coastal regions in Eurasia and Alaska are not defined as permafrost? Is it due to a lack of spatial resolution?

Authors response: JULES is missing these cells in the future projections and thus, not added to the figure. No changes made Figure 4: The Y-axis ticks for ORCHIDEE should be changed to the same as other sub-figures. Authors response: Thanks for pointing that out, now all axes in figure 4 are identical Table 3: P-values or significance tests should be addressed for these correlation coefficients. Authors response: We did ran significant tests for these correlations but did not added them. Changes: We added the stats to the figure and included the following statement in the footnote: "Pearson correlations (r) significant at *p<0.01 and **p<2e-16. "

Discussion section: In my opinion, section 4.1 and 4.2 should switch. As section 4.2 is more closely related and more important to the intercomparison results. Section 4.1, on the other hand, discusses the feature most involved models have not supported yet.

Authors response: We kept the "Permafrost degradation and drying" section first and details of uncertainty second given that this manuscript is a qualitative analysis of the trends and the causes of the trends (i.e. permafrost thaw and drying across all models). No changes were made.

Please also note the supplement to this comment:
https://www.the-cryosphere-discuss.net/tc-2019-144/tc-2019-144-AC1-supplement.pdf

**Supplement:**

[revised manuscript text omitted]

---

## Author Comment (AC2) · 27 Aug 2019

Dear Reviewer, I sincerely appreciate taking the time to review this paper and provide very helpful comments and suggestions that significantly improved the clarity, flow and message of the manuscript. I addressed every comment you had and responses are below. Tracked changes are in the supplement pdf file. FYI- modified figures in pdf will have both versions where top figures will be the old version and bottom figures will be the new, corrected version. On behalf of all authors, Thank you. Christian Andresen

Major points Reviewer #2 Abstract. The last sentence is quite general and states things that are very well known already. Could the abstract instead finish with a more inter-

esting statement pointing out specific knowledge gaps or recommended directions of research? Authors response: We agree and rewrote the last sentence of the abstract following your suggestion. Sentence now reads: "Coordinated efforts to address the ongoing challenges presented in this study will help reduce uncertainty in our capability to predict the future Arctic hydrological state and associated land-atmosphere biogeochemical processes across spatial and temporal scales".

106 Although method specifics can (hopefully) be obtained in the cited papers, I'd like a few more details here, for clarity. For one thing, it's not clear when the break point between historical, model-specific climate forcing and the common forcing took place. Was this at 1960 or at 2006? Authors response: We clarified the methods as suggested to include this detail: L107 "simulations were conducted from 1960 to 2299, partitioned by an historic (1960-2009) and future simulation (2010-2299)". 114-117 Along the same lines, for clarity here: On what timescale did the historicalCCSM4 climate forcing repeat? Authors response: That was specific for each modeling group and addressed in McGuire et al 2018 (cited in manuscript). 134 Just to be clear, specify what years of model simulations were used for the comparison with 1970-1999 observations. I assume this is also long-term but is it the exact same period, or some other length? Authors response: We used the same years of simulations for comparison and highlighted it in the footnote of Figure 6 and 7. Changes: Figure 6. Runoff anomaly comparison between gauge data and models simulations for the period 1970-1999 mean. 157 Here the authors refer to the "permafrost domain", but this is not clearly defined in methods. Please clarify in the methods sections whether the study domain is, for each model, all cells with near-surface permafrost above 45 degrees N, as suggested on lines 121-123, or something else. Authors response: This certainly needed clarification in the manuscript and we added/modified the following statements: Changes: In the first paragraph of methods section 2.2 we added: L121-123 "This qualitative hydrology comparison was based on the full permafrost domain in each model rather than a common subset among models in order to fully portray the overall changes in permafrost hydrology for participating models." We also clarified the permafrost esti-

mation for the top 3m soil column which is slightly different among models due to its soil configuration layers ranging from 2-3m. Line 123-125 now reads: "we define a grid cell as containing near-surface permafrost if the annual monthly maximum active layer thickness (ALT) is at or less than the $\sim$3m depth layer depending on the model soil configuration (Figure 1)"

168-171 I am a bit dubious as to whether these patterns hold over longer-term analysis. If this statement is supported by the comparison of 10-year averages shown in Figure3, I am unconvinced. See comment on that below. Figure 3. Here the authors use a ten-year period to illustrate long-term spatial changes. This is way too short as decadal variability is clearly substantial for some models (Figure 2c). This should be a 30-year period. Authors response: We agree, a 30-year period comparison will be more representative and strengthen the paper. Changes: We changed the analyses to 30 year averages and modified the figure 3 and 4 as suggested. No major changes were observed.

191-193 I think this statement is not supported enough by the data. Either there is a relationship or not, and it would be easier to determine the likelihood of that with a simple x-y plot of the data rather than these box plots. As the authors note, the UWVIC model is not useful at all for this question due to its resolution. But for the box plots shown, I think the SIBCASA model clearly shows no tendency for more drying with ALT increase, which is not acknowledged. The statement should be modified to moderate this claim somewhat. Also, I am wondering at the use of short time periods again here, and would prefer a 30-year period comparison. Authors response: We acknowledged that this is an important point that needed work and clarification. We are aware that these relationships are not straight forward and we highlighted it in the original text after our claim (L191-192) for fairness. Original text reads: L192-195 "However, there is a large spread between soil moisture and ALT changes (Figure 4) which may be influenced by many interacting factors that can be difficult to assess directly and are out of the scope of this study." In addition, the reason why we did not

use simple x-y plots was because boxplots were a clearer way to portray this trends and better shows the distribution of these points (compared to a scatterplot of 10,000 points). Changes: Following your suggestion, we strengthen the analysis by running the comparison analysis for a 30-year period and showed the correlation statistics for these relationships to support our statement.

222-223 According to the text, JULES exhibits the highest runoff increase with 0.8mm/day, but Figure 2g shows ORCHIDEE runoff increasing by 1.2 mm/day. Which is correct? Authors response: The statement only tries to convey that JULES has a high precipitation trend but does not imply it has the highest precipitation. No changes made

Minor and language points Authors response: We made all the changes to the document following your suggestions and edits below. 110 The degree symbol seems to have been replaced by a 0 (zero character), at least on my computer. 161 Add "long-term" or "for the period after 2100" or similar to clarify that it's only after 2100 that most models stay on the drying side for soil moisture – up till then, about half of the models are close to zero change or wetting. I guess this is implicit with the talking of 2299 in the preceding sentence but still, just to be clear. 303 Change "large-scales" to "large scales". 392 Change "Study" to "The study". Fig 1. The figure seems to show depths to 3.5 m but the caption says 3 m. Fig 2. The caption says "Figures d, e, f, and g are represented as relative change from 1960 values". I think "relative change" implies a normalization which is not done here, so I suggest dropping "relative" from the above sentence. Fig 7. At least in the pdf on my computer, the tick labels on the horizontal axis are misaligned and show up inside the plot instead of outside. Please check.

Please also note the supplement to this comment:
https://www.the-cryosphere-discuss.net/tc-2019-144/tc-2019-144-AC2-supplement.pdf

---

## Author Comment (AC3) · 16 Sep 2019

Thank you for sharing your insight on relevant hydrological processes influencing soil moisture in permafrost landscapes. Your simulations with the PWB model raise interesting questions on seasonal rainfall changes and how this will influence soil moisture and soil thermal dynamics. It is interesting to see PWB model simulations in Rawlins et al 2013 which suggests that most of the increases in precipitation will occur in the winter/spring as snow and will be lost through spring melt runoff. Most of this water will not reach the soil given that the active layer has just started to thaw in the spring. This phenomenon also could enhance soil moisture decrease during summertime. Seasonal dynamics between precipitation, runoff and evaporation is a topic that should be explored further particularly across different models. Reference: Rawlins, M.A., Nicolsky, D.J., McDonald, K.C. and Romanovsky, V.E., 2013. Simulating soil freeze/thaw dynamics with an improved panArctic water balance model. Journal of Advances in Modeling Earth Systems, 5(4), pp.659-675.

---

## Author Response (AR2)

tc-2019-144 responses to reviewers
journal article review response
en

**tc-2019-144 responses to reviewers**

**Referee #1**

I would like to thank authors to address most of my concerns, while my concern on defining permafrost region still remains in the new version of the manuscript.

The clarification on how authors define permafrost still makes me confused, if not making me even more confused than the last review. "at or less than 3 meters depending on model soil configuration" does not make the methodology any clearer to me. I apologize if I did not make myself clear in the last review. Let me clarify my concern below.

Let's assume an idealized situation. For a certain location (grid point), the ALT for CLM is 3 meters. This location is then counted as in the permafrost region for CLM. Meanwhile, for the same location (or the nearest grid point) in JULES, the soil temperatures for the top 2.8 meters (because for JULES it only has 2.8 meters of soil, according to your Figure 1) in JULES is exactly the same as those in CLM. In this case, is the same location defined as permafrost in JULES? I feel like authors would define "no permafrost" in this case according to the algorithm, but I found no clear evidence in the manuscript.

The confusion above follows me to another question—when searching for the permafrost region, do authors make the rule that "at least one soil layer should be frozen (temperature below freezing) for a grid point? I feel like authors do have done something like it, or in the idealized example I just took above, JULES will have permafrost for that grid point. On the other hand, if authors do have such a rule in searching for permafrost grids, it means the maximum active layer depth in some models are far less than in other models. For JULES it is 2.6 meters. For CoLM, it is 2.3 meters. Then it is 2.0 meters for TEM and 0.4 meters for UWVIC. I can expect the above from Figure 3 in which the area of permafrost for CoLM, TEM, and JULES is smaller than those in the upper panel. But for UWVIC, it has the biggest area of permafrost. To me, it is a little bit crazy to have a 0.4-meter ALT for Anchorage, Alaska in any kind of land model. If it is the case, maybe it is a better idea to just exclude UWVIC from inter-comparison.

I understand that defining permafrost should be in different ways for different models because of the varied soil configurations. But authors should show more details to prevent any confusion from readers.

Dear Referee,

Thank you for your comment, this is certainly a challenge when comparing multiple models with different soil configurations. We are aware of these differences in the soil column configurations, and thus, permafrost extents, therefore we clarified and highlighted in the first sentence of the methods section 2.2 and the first paragraph of the discussion that this manuscript is a qualitative analysis and does not focus on the details of magnitude and spatial patterns of the models signatures. In addition, the range of hydrologic responses in the models are broad regardless of slight differences in permafrost extent, indicating high structural uncertainty across models with respect to this particular aspect of the Arctic system response to global climate change.

Also, we are aware that UWVIC has a smaller number of soil layers which may have influenced the distribution of permafrost. However, because this model does simulate permafrost and hydrology, we decided to still include it in the manuscript and give readers a broader perspective of the current diversity of permafrost simulations by various modeling groups.

Referee #2

Comments on revised version of manuscript tc-2019-144

I thank the authors for making several substantial changes that have improved the manuscript. In particular, the change from 10- to 30-year periods and the addition of statistical tests for the ALT vs soil moisture change have reinforced the confidence of the findings the authors present. I still have a minor comment on the statistics though (see below).

Furthermore, I still haven't been able to find enough detail explaining how climate model projections were calculated. References to earlier publications are not enough to show this. See below for that issue. Line numbers below refer to the version without track changes.

Scientific points

I asked in the previous review for clarification regarding the periods for which the historical forcing was repeated, and the authors referred in their response to the McGuire 2018 paper. I am well aware of that paper, as it was cited in the original manuscript, but I still think this information should go into this manuscript, as I asked in my original question. It is important to understand how the future projections were constructed, something that is presently not clear.

The reference to McGuire 2018 is not very helpful, as that paper's methods section also does not include any detail on the repeating periods of the early 20th century forcing. There is a single sentence stating in principle the same thing as in the present manuscript, but without any further detail: "All models were driven with a common projection period forcing by applying monthly climate anomalies/scale factors from a CCSM4 simulation that included the RCP4.5 and RCP8.5 (2006–2100) and the extended concentration pathways (ECP4.5 and ECP8.5, 2101–2299) on top of repeating early 20th century reanalysis forcing". Also, this sentence is confusing, as it talks specifically about "reanalysis" datasets, unlike the present manuscript, which talks about "forcing" and "driving" datasets, the latter of which are not all reanalysis datasets. My interpretation is that the sentence in McGuire 2018 refers to the historical forcing datasets that are mentioned later in that papers' methods section, where Table 3 in McGuire 2016 is indicated for details.

Unfortunately, Table 3 in McGuire 2016 is also not helpful to understand the repeating periods. It does state the historical forcing dataset names, but the time periods are given only for some of the models. In any case, since the 2016 paper does not involve future projections at all, it therefore doesn't involve any repeating of historical forcing. So – unless I have missed something – it is not possible from either McGuire 2016 or 2018 to know the periods of the different repeating historic forcing atmospheric datasets with which the common CCSM4 future projections were compared.

I find this problematic, but it should be possible to now to either add this information, for example by joining it to the listing of included historical forcing datasets in Table 1, or in a supplement, or at least explain more clearly what was done. This would substantially help interpreting how the CCSM4 future projections were calculated, both in this paper and in McGuire 2018.

On a general note, the approach used here introduces a risk of bias both due to the use of a single projections model, which could be skewed towards the high or low range of the range of climate model responses to RCP forcing, and also due to projections being compared to different historical baselines. As for the model choice, I understand this is based on the model being fit-for-purpose in terms of the high-latitude water cycle, as explained and shown in McGuire 2018, so I don't have an issue with the specific choice of model – I just think this choice, and the choice to compare future changes that are measured against different baselines, should be motivated. The authors both in the paper and in their responses refer to previous publications describing this, but I think they could afford to spend a few lines motivating these key choices and discussing their possible influence on their results also in the present paper. The methods section as it stands is very brief (about 550 words).

Thank you for your feedback, we added the repeating periods to the methods and addressed the motivation of our key choices for the methods section in the following paragraphs:

(115-121) *"Future simulations were calculated from monthly CCSM4 (Gent et al., 2011) climate anomalies for the Representative Concentration Pathway (RCP 8.5, 2006-2100) and the Extension Concentration Pathway (ECP 8.5, 2101-2299) scenarios, relative to repeating (1996-2005) forcing atmospheric datasets from the different modeling groups (Table 1)."*

(122-135) *"The choice of the PCN model intercomparison was to use output from a single Earth System model climate projection was motivated by a desire to keep the experimental design simple and computationally tractable. Clearly, using just one climate projection does not allow us to explore the impact of the broad range of potential climate outcomes that are seen across the CMIP5 models. Instead, the PCN suite of simulations allows for a relatively controlled analysis of the spread of model responses to a single representative climate trajectory. The selection of CCSM4 as the climate projection model was motivated partly by convenience and also because it was one of the only models that had been run out to the year 2300 at the time of the PCN experiments. Further, as noted in McGuire et al. (2018), CCSM4 late 20th century climate biases in the Arctic were among the lowest across the CMIP5 model archive. It should be noted that the use of a single climate projection means that the results presented here should be viewed as indicative of just one possible permafrost hydrologic trajectory. As we will show, even under this single climate trajectory, the range of hydrologic responses in the models are broad, indicating high structural uncertainty across models with respect to this particular aspect of the Arctic system response to global climate change."*

137-143 The description of model-observation comparison for runoff is incomplete and ambiguous. From the results, it's clear the authors did three things: 1) compared the pattern of modeled and observed annual discharge values for 1970-1999 (visually, Fig 6), 2) determined the correlation between modeled and observed annual discharge values (Table 3), and 3) compared the distributions of modeled and observed annual discharge values for 1970-1999 (Figure 7). The methods should describe the things the authors actually did – the present two sentences "We compared model simulations with long-term (1970-1999) mean monthly discharge data from Dai et al 2009. We computed model mean annual discharge including surface and subsurface runoff for the main river basins..." do not do this. The first sentence could just as well mean that the authors did a model-observation comparison on the long-term monthly climatology.

*We strengthen the methods section on runoff model-observation comparison description and added the main analysis shown in the results. The section now reads: "We computed model total annual discharge (sum of surface and subsurface runoff) for the main river basins in the permafrost region of North America (Mackenzie, Yukon) and Russia (Yenisei, Lena). In particular, we compared (i) annual runoff anomalies, (ii) correlation coefficients and (iii) distributions of annual discharge between gauge data and models' simulations for the 30-year period of 1970-1999. "*

Figure 4
The figure is now clipped for some of the models so that the box plots are only partly visible; please correct.
Statistics – the Pearson r is presented. Typically Pearson r denotes a sample correlation coefficient, while Pearson rho denotes a population correlation coefficient. To me it makes sense to use the population version here, as we are not looking to estimate a population correlation from a sample, but rather trying to understand the correlation in this particular set of paired points, which should be thought of as the entire population. Of course, with 10,000 data pairs it will not make a difference, but I think the notation should be correct and correspond to the formula used to really calculate the coefficient (whether it is the sample or population correlation should be stated in the Figure caption).

*Figures were corrected to show full boxplots as suggested. Also, we clarified figure 4 adding that Pearson correlations where population correlations. While running the code for figure 4, we realize we had a bug for UWVIC boxplot. The new figure reflects the correct trend for UWVIC.*

Section 3.3 numbers – in this section, the authors present mean numbers and a range for several different quantities. Please clarify in the text what the mean and range refer to – for example, if it is standard deviation, and between what values in that case.

*We clarified text by adding : "….increase is 0.1±0.1mm/day (mean ± SD, hereafter)……."*

229-230 The statement about JULES runoff is still problematic for the same reason I originally pointed out. In their response, the authors mention precipitation and that they made no changes, but the statement I am talking about refers to runoff. I ask the authors to change this statement for clarity. The interpretation of lines 229-230 reads as JULES having the highest runoff values of all models, which is not correct. If the authors want to convey that the high runoff and precipitation changes in JULES are consistent with each other, they should say so clearly and not focus on the JULES runoff value as being the highest. If they want to mention the models at the high end of the runoff projection range (which seems more likely, given the context of the preceding sentences), they should mention that both JULES and ORCHIDEE are far above the 0.2 to 0.3 mm/day range that they mentioned in the preceding sentence, and not state that JULES has the highest value, as the one for ORCHIDEE is higher.

*Thank you for pointing that out, we deleted JULES (229-230) sentence to avoid confusion.*

Minor remarks
76-78 This sentence talks about examples of model upgrades to "soil thermal dynamics and active layer hydrology", but the last example is termed simply "cold region hydrology". This seems a bit backwards to me – soil thermal dynamics and active layer hydrology are subsets of cold region hydrology, not the other way around. I suggest this should be rephrased to be accurate.

80-81 "models simulations", correct plural forms.

*Corrected to : "models' simulations"*

Replace "forced with a common projected climate" with "the latter forced with a common projected climate", to clarify that it is only the future period that has a common forcing.

*Changed as suggested. To clarify the sentence, we added: "where the future simulation was forced with a common projected climate"*

107-108 Similarly, I think "historic (1960-2009) and future simulations (2010-2299)" works better than the presently written "an historic (1960-2009) and future simulation (2010-2299)", as the simulations are not strictly the same but differ between models.

*Changed as suggested, deleted "an"*

Related to the major point about the repeating historical forcing, the phrase "overlaid by repeating historic forcing atmospheric datasets from CCSM4" sounds odd. The historic forcing atmospheric datasets were not from CCSM4, only the future climate. I suggest to rephrase this sentence to something like "Future simulations were calculated from monthly CCSM4 (Gent et al., 2011) climate anomalies for the Representative Concentration Pathway (RCP 8.5, 2006-2100) and the Extension Concentration Pathway (ECP 8.5, 2101-2299) scenarios, relative to repeating historic forcing atmospheric datasets from the different modeling groups (Table 1)."

*Changed as suggested*

I would call this a correlation or association, not a trend. As terms go, "trend" works better to denote a rate of change over time, but this sentence refers to a spatial association that is not analyzed with respect to time.

*Changed as suggested, "trend" to "correlation"*

I suggest putting "except SIBCASA, LPJGUESS and UWVIC" in parentheses rather than between commas to avoid any ambiguity that these models are the exception from the ALT increase-soil moisture decrease relation.

*Changed as suggested, we put in parentheses: (except SIBCASA, LPJGUESS and UWVIC)*

Remove "mean" from the end of the figure caption for Figure 6.

*We removed "mean"*

[revised manuscript text omitted]

---

## Author Response (AR3)

Editor Minor comments on Manuscript  TC-2019-144
Changes of manuscript are marked in the pages below

You appear to have missed a minor comment from Reviewer 2 on lines 76-78.
Changed "cold region hydrology (Swenson et al., 2012)" to "*hydraulic properties of frozen soils (Swenson et al., 2012)*" to be more specific
- Please check the formulation of the first sentence of the new paragraph starting on line 119. (Perhaps the language is as you intend, but it seems somewhat off to me.)
First sentence now reads: *The PCN model intercomparison uses the output from a single Earth System model climate projection and was motivated by a desire to keep the experimental design simple and computationally tractable.*

- Check the reference on line 151. (Dai et al 2009, should be Dai et al. (2009)?)
Changed as suggested

Most importantly, the main concern of Reviewer 1 has still not been addressed in the manuscript. Please make sure that there is a clear description of how permafrost was defined across models.
We clarified this in the text a bit better by adding a few sentences that address the specific calculations. Text now reads:

[revised manuscript text omitted]